# Study on the Extraction Technology and Antioxidant Capacity of *Rhodymenia intricata* Polysaccharides

**DOI:** 10.3390/foods13233964

**Published:** 2024-12-09

**Authors:** Shiyi Dong, Yu Wu, Yutong Luo, Wanxia Lv, Shuyi Chen, Nannan Wang, Meihan Meng, Ke Liao, Yichao Yang

**Affiliations:** 1School of Public Health, Guangzhou Medical University, Xinzao, Panyu District, Guangzhou 511436, China; ddssyy1111@163.com (S.D.);; 2Public Technical Service Center, Guangzhou National Laboratory, Guangzhou 510005, China

**Keywords:** *Rhodophyta intricata*, ultrasonic-assisted extraction, response surface design, antioxidant activity

## Abstract

A red alga named *Rhodymenia intricata* was explored, and the extraction technology and antioxidant capacity of its polysaccharides were investigated. The crude polysaccharides were extracted using the ultrasound-assisted water extraction method, precipitated by alcohol, and purified using the trichloroacetic acid method. Subsequently, the scavenging rates of polysaccharides on hydroxyl, *DPPH*, and *ABTS* free radicals, were determined both prior to and following purification to evaluate their antioxidant activity. Extraction technology was optimized to improve polysaccharide yield, and the optimal parameters were as follows: particle size 100 mesh, material–liquid ratio 1:84 (g/mL), ultrasonic time 30 min, and extraction for 95 min at 80 °C. The maximized extraction rate of crude polysaccharides was 37.78 ± 0.15%. The obtained crude polysaccharides were purified with different concentrations of trichloroacetic acid, and the purification effect was evaluated according to protein removal rate and polysaccharide retention rate, which could reach 62.61 ± 1.82% and 96.10 ± 1.60%, respectively. Infrared spectrum analysis suggested that *Rhodymenia intricata* polysaccharide might be α-pyranose. The Congo red test illustrated that the polysaccharide contained a triple helix structure. In the antioxidant activity assessment, the scavenging rates of polysaccharide prior to purification for RIP-1 (10 mg/mL) for hydroxyl, *DPPH*, and *ABTS* free radicals were observed to achieve maximum values of 94.71 ± 0.13%, 42.80 ± 7.12%, and 76.30 ± 5.20%, respectively. In contrast, the scavenging rates of polysaccharide following purification for RIP-2 (10 mg/mL) for the same free radicals reached maximum values of 94.10 ± 0.27%, 32.37 ± 0.78%, and 98.30 ± 0.34%, respectively. Notably, these scavenging rates exhibited a dose-dependent relationship. These results demonstrated the potential of the extraction method for polysaccharides from *Rhodymenia intricata,* and for adding value to the by-product for its potential application as an antioxidant in food and pharmaceutical products.

## 1. Introduction

There are abundant marine resources in China, among which macroalgae are a significant component. According to statistics, there are up to 1277 recorded species of seaweed along the Chinese coast, with 607 belonging to the *Rhodophyta division* alone [1]. Unlike terrestrial plants and animals, the cell wall matrix of *Rhodophyta division* contains a large amount of polysaccharides, which exhibit unique structural and biological characteristics, such as antioxidant [2], anti-microbial [3], anti-inflammatory [4], anti-tumor [5], prebiotic [6], immunomodulatory [7], and hypoglycemic properties [8], among others. These properties make polysaccharides from *Rhodophyta division* a highly favored source of food supplements, functional foods, and nutritional healthcare products [9]. They are one of the research hotspots in the fields of food and biomedical science. *Rhodophyta division* is rich in polysaccharides (360~660 g·kg^−1^ (by dry weight)) which possess various potential types of bioactivity. For example, the sulfated galactans (SGs) from *S. hypnoides* exhibited potential dose-dependent antioxidant activity scavenging free radicals, such as hydroxyl (65.58% at 3 mg/mL), *DPPH* (56.41% at 2 mg/mL), and superoxide radicals (73.12% at 0.6 mg/mL) [10]. This is exactly the reason that marine macroalgae could be the main source of bioactive compounds, which tips us off to the important value of polysaccharides from *Rhodophyta division* in the prevention and treatment of disease [11]. Therefore, the extraction of bioactive polysaccharides from *Rhodophyta division* possesses sustainable resource advantages and holds significant research value.

To realize the intensive development of *Rhodophyta division*, extracting bioactive polysaccharides from *Rhodophyta division* is crucial. *Rhodymenia intricata* is a marine economic red alga belonging to the genus *Rhodymenia* within the family *Rhodymeniaceae*. The algae of *Rhodymenia intricata* are characterized by their bright red coloration and membranous structure. The reproductive capsules are either spherical or hemispherical and develop along the edges of the branches. This species typically inhabits rocky substrates within the low tide zone or subtidal regions, thriving on reefs and rocky swamps. It is distributed in Japan, South Korea, and the eastern coastal areas of China. It is classified as a warm temperate seaweed. Through testing conducted by an independent third-party institution, it was found that the carbohydrate content of *Rhodymenia intricata* accounts for nearly half of its total composition. Its polysaccharide content significantly exceeds that of other red alga species. Therefore, utilizing it as a raw material for polysaccharide extraction holds vast potential for application.

The structure and activity of polysaccharides are closely related, which in turn are influenced by their extraction conditions, including the composition of the monosaccharides, their molecular weights, and respective distributions [12]. Currently, the most commonly used polysaccharide extraction methods in research include solvent extraction [13], ultrasonic [14], and microwave or enzyme-assisted extraction [15]. Commonly used solvents in extraction methods include water, alkaline solutions, and acidic solutions. Given that polysaccharides are soluble in water but insoluble in ethanol, water is employed for their extraction, followed by precipitation using ethanol to separate the polysaccharides from the solution. This method is regarded as safe and cost-effective and has been widely utilized in research focused on the extraction of plant polysaccharides. However, alkaline extraction of polysaccharides is susceptible to hydrolysis and typically requires protective measures, such as the use of nitrogen or the addition of sodium borohydride, to mitigate degradation. Besides, alkaline extraction is primarily suitable for raw materials characterized by low pectin content and low viscosity. Acidic extraction can also lead to the cleavage of glycosidic bonds and degradation of polysaccharides, necessitating careful selection of experimental conditions. Furthermore, ultrasonic-assisted extraction exploits the ability of polar molecules within cells to absorb ultrasonic wavelengths, resulting in mechanical stress that generates heat and causes cell rupture. This process facilitates the release of intracellular components, thereby increasing yield. Compared to conventional extraction methods, ultrasonic-assisted extraction integrates ultrasonics with traditional solvent techniques, demonstrating significant improvements in extraction efficiency through enhanced yield and reduced extraction time [16]. It should be noted that different extraction methods seriously affect the yield and biological activity of polysaccharides, and selecting the appropriate extraction method is crucial for polysaccharide preparation.

Univariate analysis was applied to investigate the influences of a single condition on polysaccharide yield and molecular structure, and the interactions among factors were also analyzed to find the optimal extraction conditions. Furthermore, extracted polysaccharides were purified using trichloroacetic acid (TCA), and their antioxidant activity in vitro was evaluated. Accordingly, this study can provide valid experimental data and theoretical support for the further development and utilization of polysaccharides, promoting the high-value application of polysaccharides from *Rhodymenia intricata*.

## 2. Materials and Methods

### 2.1. Materials and Reagents

The *Rhodymenia intricata* was obtained from Zaochen Aquatic Products Co., Ltd., Dongshan County, Zhangzhou City, Fujian Province, China. The hydroxyl radical assay kit was acquired from Nanjing Jiancheng Bioengineering Institute Co., Ltd. (Nanjing, China), and other reagents were purchased locally.

### 2.2. Optimization of the Polysaccharide Extraction Process

#### 2.2.1. Preparation of Crude Polysaccharides

The *Rhodymenia intricata* was freeze-dried using FD-1A-80 vacuum freeze dryer (Beijing Boyikang Laboratory Instrument Co., Ltd., Beijing, China) and ground into powder. After degreasing, drying, and sieving, the *Rhodymenia intricata* samples were obtained. A 1.00 g *Rhodymenia intricata* sample (*M_s_*) was put into a beaker and accurately weighed, and deionized water was added according to the predetermined solid-to-liquid ratio, thoroughly stirred, and placed in an ultrasonic disintegrator for treatment (5 min, 330 W). After the disruption, it was placed in an ultrasonic cleaner for homogenization, and then transferred to a water bath pot for a period in a constant temperature water bath. After centrifuging for 10 min at 5000 r/min, the precipitate was washed with anhydrous ethanol and dried to acquire crude polysaccharides of *Rhodymenia intricate* (*M_p_*). This experiment was repeated in triplicate. Polysaccharide yield could be obtained using the formula:(1)$$\mathrm{Extraction}\mathrm{yield}\;(\%)=\frac{M_{p}}{M_{s}}\times 100$$

#### 2.2.2. Single-Factor Experiment

According to the above-described polysaccharide extraction method, effects of different factors on extraction yield were evaluated by single-factor experiments. In order to obtain the optimized experimental conditions, each parameter was systematically varied at five levels: the material-liquid ratio (g/mL) was set at 50, 75, 100, 125, and 150; extraction temperature was 60, 70, 80, 90, and 100 °C; extraction time was 50, 75, 100, 125, and 150 min; the particle size after grinding was 40, 60, 80, 100, and 120 mesh; and ultrasonic time was 20, 30, 40, 50, and 60 min.

#### 2.2.3. The Steepest Climbing Experiment

To accelerate the parameter optimization process, the step size and direction of the steepest climbing experiment were optimized according to the individual factor effects in Table 1. This experiment had three replicates.

#### 2.2.4. Design of the Response Surface Methodology

The above five independent factors can be further studied in Table 2, using a Box–Behnken design. A total of 46 experiments were included in the Box–Behnken design and implemented in the order shown in Table 2 to assess synergistic effects between variables and responses. All experiments were repeated three times.

### 2.3. Purification of Polysaccharides from Rhodymenia intricata

Polysaccharides were extracted under the optimized conditions. RIP-1 could be dissolved in a 3 mg/mL polysaccharide solution of *Rhodymenia intricate*. The polysaccharide content (*M*) and protein content (*M*′) of the solution were measured before deproteinization. Then, the 3 mg/mL polysaccharide solution was mixed with 4%, 6%, 8%, 10%, and 12% TCA solutions at a ratio of 4:1 (VRIP-1/VTCA), shaken for 20 min, and stored at 4 °C overnight. After being centrifuged for 15 min at 5000 rpm, polysaccharide content (*m*) and protein content (*m*′) in the supernatant were measured after deproteinization. The polysaccharide retention rate (*U*) and the protein removal rate (*B*) were computed according to the following formula:(2)$$U\left(\%\right)=\frac{M-m}{M}\times 100$$
(3)$$B\left(\%\right)=\frac{M^{\prime }-m^{\prime }}{M^{\prime }}\times 100$$

A comprehensive score was received on the basis of the means of the polysaccharide retention rate and protein removal rate. All trials were performed in triplicate, as follows: collect the supernatant, dialyze for 48 h, concentrate under reduced pressure to a certain volume, and freeze-dry, thus obtaining the freeze-dried sample of deproteinized *Rhodymenia intricate*, designated as RIP-2.

### 2.4. Determination of Physicochemical Properties

Total sugar and protein content was examined using the method of Yue [17] and the Bradford method [18]. The sulfate and total phenolic content was measured using the Hartiala [19] and the Folin–Ciocalteu method [20], respectively.

### 2.5. Structural Analysis

#### 2.5.1. Fourier Transform Infrared Spectroscopy (FT-IR) Analysis

An FT-IR spectrometer (PerkinElmer Frontier, Paris, France) was utilized to acquire the FT-IR spectrum of 400–4000 cm^−1^.

#### 2.5.2. Congo Red Test

The Congo red test was performed by Nie [21]. Firstly, spectral scanning within the wavelength range of 400~600 nm was carried out to find the maximum absorption wavelength (*λ_max_*) using a UV–Vis Spectrophotometer (Hitachi, Tokyo, Japan). *λ_max_* was on the vertical axis, and NaOH concentration was on the horizontal axis of a plot fitting a curve. Pure water was used as the negative control.

#### 2.5.3. Analysis of the Antioxidant Property

The antioxidant capacity of polysaccharides was evaluated from the scavenging activity of the hydroxyl radical, *DPPH* free radical, and *ABTS* free radical, examined using their respective assay kits. The concentration of polysaccharide solutions was 2, 4, 6, 8, and 10 mg/mL, and vitamin C was employed as the positive control. Statistical analysis was conducted and plotted using Origin 2020 software (OriginLab, USA). Scavenging activity of the hydroxyl radical, *DPPH* free radical, and *ABTS* free radical could be obtained using the following formulas:(4)$$OH\;scavenging\;rate\;\left(\%\right)=\frac{A_{control}-A_{sample}}{A_{control}-A_{blank}}\times 100$$
(5)$$DPPH\;scavenging\;rate\;(\%)=\frac{{1-(A}_{sample}-A_{control})}{A_{blank}}\times 100$$
(6)$$ABTS\;scavenging\;rate\;(\%)=\frac{{A_{blank}-A}_{sample}}{A_{blank}}\times 100$$

## 3. Results

### 3.1. Improvement of the Extraction Technology of Polysaccharide

#### 3.1.1. Single-Factor Analysis

The concentration of polysaccharides from *Rhodymenia intricata* in solution is affected by the material–liquid ratio. Both too low and too high concentrations can impact the extraction yield. As shown in Figure 1A, polysaccharide yield kept rising with the increase of the material–liquid ratio in the initial stage. When it reached 75 g/mL, the polysaccharide extraction yield was the highest. Beyond this point, as the material–liquid ratio increased, polysaccharide yield decreased. This could be due to using too little material compared to the solvent, which increases the viscosity of the polysaccharides and make them hard to dissolve, resulting in a lower extraction yield. An excessively high ratio of solvent to material may reduce or disperse the effect of ultrasound on *Rhodymenia intricata* and also increases the difficulty of the extraction process, causing polysaccharide loss and leading to the decrease in extraction yield [22].

The dissolution rate of polysaccharides of *Rhodymenia intricata* is affected by extraction temperature, and excessively low temperature can prolong the time required for complete dissolution of the polysaccharides in solution, while excessively high temperature can damage the polysaccharide structure and reduce polysaccharide yield. As the temperature was elevated, the solubility of polysaccharides in water gradually increased, and the rate of molecular motion increased, accelerating the interaction between the solute and solvent, with the highest yield achieved at a temperature of 80 °C (Figure 1B). However, an excessively high temperature can cause polysaccharide degradation and glycosidic bond cleavage, damaging the polysaccharide structure and decreasing extraction yield [23]. Consequently, 80 °C was optimal for the preparation of polysaccharides from *Rhodymenia intricata*.

The degree of dissolution of polysaccharides from *Rhodymenia intricata* in solvent is affected by extraction time. An excessively short extraction time results in incomplete dissolution, while an excessively long time may damage the polysaccharide structure. Extraction yield increases with the extraction time, and increases when extraction time is insufficient (Figure 1C). However, after 100 min, the extraction yield decreased. This may be because polysaccharides require time to precipitate, and an excessively short extraction time may result in incomplete extraction or dissolution of polysaccharides. Some polysaccharides remained undissolved in the solution, making it impossible to extract them further in subsequent experiments. Too long a time, on the other hand, can damage the polysaccharide structure, leading to the decrease in extraction yield [24]. Thus, an extraction time of 100 min was optimal.

The amount of polysaccharide extracted is affected by the particle size of the crushed *Rhodymenia intricata*. *Rhodymenia intricata* with a smaller mesh size (larger particle size) is less likely to release polysaccharides. Moreover, *Rhodymenia intricata* with a larger mesh size (smaller particle size) is prone to agglomeration, which affects the polysaccharide extraction yield. Polysaccharide yield first increased and then decreased as the mesh size increased, reaching a maximum at a mesh size of 100 (Figure 1D). This may be due to the higher degree of crushing of *Rhodymenia intricata*, which enhances cell wall disruption, allowing polysaccharides in the cytoplasm to dissolve and bind with more water molecules. However, the polysaccharide extraction yield decreased sharply after a mesh size of 100, possibly because an excessively fine particle size (large mesh number) made the *Rhodymenia intricata* material prone to agglomeration and settlement in aqueous solution, which was unfavorable for polysaccharide dissolution.

As shown in Figure 1E, when the ultrasonic time was less than 30 min, increasing ultrasonic time led to a gradual increase of polysaccharide extraction yield. This may be due to the improved homogenization effect with increased ultrasonic time, resulting in more polysaccharides dissolving into the extract. However, when the ultrasonic time exceeded 30 min, increasing ultrasonic time made the polysaccharide yield decrease. The reason might be that, under the action of intense ultrasound, exceeding a certain ultrasonic time can cause partial destruction of the polysaccharide structure, leading to a decrease in the polysaccharide extraction yield [25]. Therefore, selecting an ultrasonic time of 30 min was more appropriate.

#### 3.1.2. The Steepest Ascent Experiment for Polysaccharide Extraction Yield

According to single-factor experiments, the above five factors were considered to have the greatest impact on polysaccharide preparation. The central values of the three-step hill-climbing experiment were material–liquid ratio 75 g/mL, extraction temperature 80 °C, extraction time 100 min, particle size after grinding 100 mesh, and ultrasonic time 30 min. The step length was determined based on horizontal spacing. Extraction yields for the second step were the highest (Table 1) in the hill-ascending experiment.

The highest extraction yield was achieved at the second-step experiment level, and thus the factor was used as the focal point (Table 2).

#### 3.1.3. Optimization of Extraction Conditions

The extraction conditions were further optimized utilizing response surface analysis of a five-factor and three-level experiment. The matrix design and the corresponding results were summarized in Table 3.

Design Expert^®^ software 10 software (Stat-Ease, Inc., Minneapolis, MI, USA) was utilized to conduct regression analysis of polysaccharide extraction yield, obtaining the following fitted equation:(7)$$\begin{array}{cc} Y\left(\%\right)=37.2483 & +\;2.26313A+0.001875B-1.10875C+1.54D-1.06375E\\ & -\;1.055AB-2.48AC+2.035AD+1.1525AE-2.1725BC\\ & -\;0.185BD+1.2BE+0.495CD-3.7975CE+0.25DE\\ & -\;4.20229A^{2}-3.39896B^{2}-5.44812C^{2}-5.97646D^{2}\\ & -\;3.13312E^{2} \end{array}$$

ANOVA was employed to resolve the interactions among the five factors and develop a linear fit model (Table 4). The A-A model with *p*-value < 0.0001 indicated a great effect, while in the D-D model it was found that the fit index was not of significance, with *p*-value 0.0596. Additionally, the multiple correlation coefficient (R^2^) was 88.25%, suggesting that the variance was considered good and the data adequately fitted the model, with R_Adj_^2^ being 0.7885. Taken as a whole, the model demonstrated a high level of significance, indicating its applicability, and possessed good precision and accuracy. A larger *F*-value in Table 4 indicates a stronger influence of the factor on polysaccharide yield.

By comparing the *F*-values of each factor, the effects of the five factors on polysaccharide yield was: A > D > C > E > B. The lower *p* was, the more significant the corresponding coefficient would be. The linear coefficients A and D, the interaction CE, and the quadratic coefficients A^2^, B^2^, C^2^, D^2^, and E^2^ had significant effects on polysaccharide yield (*p* < 0.05) (Table 4). In addition, the linear coefficients C and E and the interaction terms AC and BC had a significant effect on the extraction rate of crude polysaccharides (*p* < 0.01). Response surface and 3D contour plots were developed to visualize the interaction between two factors (Figure 2). The figures visually illustrate the interaction between the two test variables. In Figure 2B, when other factors are held constant at zero, the polysaccharide yield initially increases with the material–liquid ratio and time, before experiencing a slight decline. Three-dimensional plots and contour plots of extraction time and temperature are presented in Figure 2E. When all other factors remain at the zero level, the variation trend of polysaccharide yield in relation to extraction time and extraction temperature resembles that observed previously. Additionally, the response surface exhibits a relatively steep gradient, indicating a strong interaction among the aforementioned factors.

According to the prediction results of the regression model, the optimized levels were as follows: material-liquid ratio 1:84 (g/mL), temperature 80 °C, extraction time 95 min, particle size 100 mesh, and ultrasound time 30 min. The predicted extraction yield of polysaccharides is 37.90% under the optimal conditions. A verification experiment was also performed based on the above results to further validate the accuracy of the model equation. The mean ± standard deviation of the results from three parallel experiments was 37.78 ± 0.15%, close to the predicted value, indicating that the stability of the optimal conditions for extracting crude polysaccharides from *Rhodymenia intricata* was good. The findings of this work are consistent with Tan et al. [26], who employed ultrasound-assisted aqueous two-phase extraction to enhance product yield of polysaccharides from *Cornus officinalis* fruit by 7.85 ± 0.09%. These results indicate that the cavitation effect generated by ultrasound is effective in disrupting cellular structures, facilitating the dissolution of polysaccharides, and, ultimately, increasing product yield. The ultrasound-assisted water extraction method not only enhances the efficiency of extraction but also contributes to cost savings in production.

### 3.2. Analysis of Purification Results of Polysaccharides from Rhodymenia intricata

The results of protein removal from polysaccharides of *Rhodymenia intricata* can be found in Table 5. When the concentration of TCA was 4%, more polysaccharides were retained, but the protein removal rate was the lowest, only 38.95 ± 2.91%. When the concentration increased to 8%, the removal rate reached the maximum of 62.61 ± 1.82%, with a polysaccharide retention rate of 96.10 ± 1.60%. Compared with 8%, when the concentration increased to 10% and 12%, the protein removal effect was significantly reduced. Excessively high concentrations not only reduce the removal rate but also tend to alter the conformation of polysaccharides, resulting in more polysaccharide loss. Based on comprehensive scoring, a trichloroacetic acid concentration of 8% was selected as the optimal condition for protein removal using the trichloroacetic acid method.

### 3.3. Analysis of Physicochemical Properties of Polysaccharides from Rhodymenia intricata

The calculation results are shown in Table 6. They indicate that the polysaccharides extracted under the optimized process conditions of this experiment had a high protein content. Further separation and purification were required to increase the purity of the polysaccharides, so as to clarify the structure–activity relationship of the polysaccharides in subsequent studies.

### 3.4. FT-IR Spectral Analysis

The resulting spectra of polysaccharides RIP-1 and RIP-2 from *Rhodymenia intricata* are shown in Figure 3. The broad absorption peak between 3200 and 3600 cm^−1^ may correspond to the O–H stretching vibration characteristic of polysaccharides from *Rhodymenia intricata* [14]. The peak at 2930 cm^−1^ may indicate the C–H stretching vibration of alkyl groups [27]. The absorption peaks at 1637 cm^−1^ and 1652 cm^−1^ may be associated with the asymmetric stretching vibration of C=O [27]. The absorption peaks at 1148 cm^−1^ and 1147 cm^−1^ may be associated with the asymmetric stretching vibrations of O–C–O and S=O, suggesting the existence of sulfate in the polysaccharides [28]. The absorption peaks at 1016 cm^−1^ and 1017 cm^−1^ may correspond to the C–O–C stretching vibration, possibly related to the pyranose ring [29]. The peaks at 833 cm^−1^ and 816 cm^−1^ may correspond to the absorption peak of α-glycosidic bonds [30]. The absorption peak of RIP-1 at 1415 cm^−1^ may correspond to the stretching vibration of the C–O bond in carboxyl groups, and the 1216 cm^−1^ peak may be attributed to the O–H bond [14]. The absorption peak of RIP-2 at 1533 cm^−1^ may correspond to the stretching vibration of the C–O or N–H bond in proteins [14]. The IR results indicated that the polysaccharide might be an α-pyranose.

### 3.5. Analysis of Congo Red Test Results

Congo red can complex with polysaccharides possessing a triple-helical structure, resulting in the rise of the characteristic peak [31]. Within a certain concentration range, this is manifested as a characteristic shift of the peak towards a violet-red color. Increasing the pH can cause the triple-helical conformation of polysaccharides to unwind into a single coiled conformation that decreases the likelihood of complexation reactions. When the concentration of NaOH exceeds a certain value, the maximum wavelength of the product drops sharply compared to the maximum absorption wavelength of Congo red [31]. As shown in Figure 4, after mixing *Rhodymenia intricata* polysaccharide RIP-2 with Congo red, its maximum absorption wavelength increased from 498.5 nm to 557 nm, indicating a significant red shift and the formation of a polysaccharide–Congo red complex in the solution. Subsequently, as the concentration of NaOH increased, the characteristic peak of the RIP-2–Congo red complex reduced quickly, and the degree of reduction was greater compared with the Congo red control group. This suggested that the triple-helical structure of the polysaccharide dissociated under alkaline conditions [32], demonstrating that RIP-2 possesses a triple-helical structure. When the NaOH concentration was 0.1 mol/L, the maximum absorption wavelength of RIP-1 underwent a red shift within the range of 495–507 nm, indicating that RIP-1 possesses a triple-helical structure. However, after the red shift of RIP-1, increased NaOH concentration did not cause the decrease of the characteristic peak of the RIP-1–Congo red complex. This polysaccharide solution may have been interfered with by multiple factors, such as metal ions and solution pH, making the helical conformation resistant to disruption under alkaline conditions [26].

### 3.6. Analysis of Antioxidant Results

To accurately understand the antioxidant activity, we need to combine the results of different antioxidant capacity tests for a thorough analysis. Polysaccharides showed scavenging activity on hydroxyl, *DPPH*, and *ABTS* in a dose-dependent manner. The results show that the free radical scavenging ability was dose-dependent and more pronounced at higher concentrations (Figure 5). In addition to this, the free radical scavenging capacity is also related to the sulfate content, and increases with the sulfate content. The same phenomenon was also found in the study of Zheng et al.; the antioxidant capacity of the polysaccharides extracted using the ultrasound-assisted method was superior to that obtained through hot water extraction [33]. This may be due to the fact that ultrasound increases the sulfate content, causing the polysaccharide to contain more active substances.

As shown in Figure 5A–C, in the antioxidant activity assessment, the lowest clearance effects of polysaccharide prior to purification for RIP-1 (2 mg/mL) for *·OH*, *DPPH*, and *ABTS* were 90.70 ± 0.68%, 32.12 ± 5.08%, and 39.23 ± 3.76%, respectively, and the lowest clearance effects of polysaccharide following purification for RIP-2 for *·OH*, *DPPH*, and *ABTS* were 80.25 ± 0.68%, 28.69 ± 1.63%, and 89.00 ± 1.42%, respectively. Meanwhile, the scavenging rates of RIP-1 (10 mg/mL) for hydroxyl, DPPH, and ABTS free radicals were observed to achieve maximum values of 94.71 ± 0.13%, 42.80 ± 7.12%, and 76.30 ± 5.20%, respectively. In contrast, the scavenging rates of RIP-2 (10 mg/mL) for the same free radicals reached maximum values of 94.10 ± 0.27%, 32.37 ± 0.78%, and 98.30 ± 0.34%, respectively. The scavenging rates against *·OH*, *DPPH*, and *ABTS* analyses showed that RIPs extracted using our method demonstrated greater antioxidant activity than *Porphyra haitanensis* polysaccharides obtained by Dong et al. [34]. In the assays for *·OH*, *DPPH*, and *ABTS* scavenging activities, VC exhibited the strongest scavenging effect. Overall, RIPs are more effective at scavenging *·OH* free radicals, followed by *ABTS*, and *DPPH* free radicals.

## 4. Conclusions

This study focused on *Rhodymenia intricata*, conducting a preliminary exploration of the extraction process and activity of its crude polysaccharides. The ultrasonic extraction process of polysaccharides from *Rhodymenia intricata* was optimized and improved utilizing single-factor experiments, the steepest ascent method, and response surface analysis. Under the optimal experimental conditions, the crude polysaccharide extraction yield was 37.78 ± 0.15%. This process is simple to operate, easy to control, and has a high polysaccharide yield, laying the foundation for the large-scale preparation of polysaccharides from *Rhodymenia intricata*. This study investigated the extraction and purification of polysaccharides from *Rhodymenia intricata*, and conducted a preliminary analysis of their antioxidant activity. This work can provide valid data and a theoretical basis for the use of *Rhodymenia intricata* polysaccharides as functional foods with antioxidant properties, and offers a reference for the prevention and adjuvant treatment of excessive free radicals in the body. However, further research is needed to explore this area.

## Figures and Tables

**Figure 1 foods-13-03964-f001:**
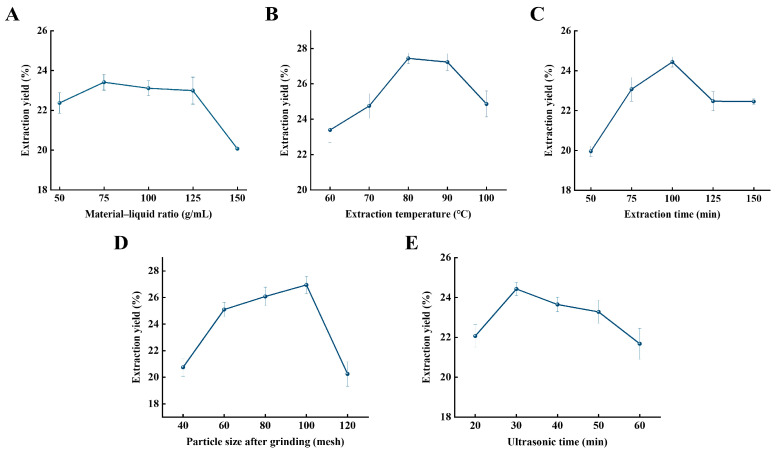
Effect of five independent factors on the polysaccharide yield of crude polysaccharide. Material–liquid (**A**), extraction temperature (**B**), extraction time (**C**), particle size after grinding (**D**), and ultrasound time (**E**) on the extraction rate of crude extract of polysaccharide, respectively.

**Figure 2 foods-13-03964-f002:**
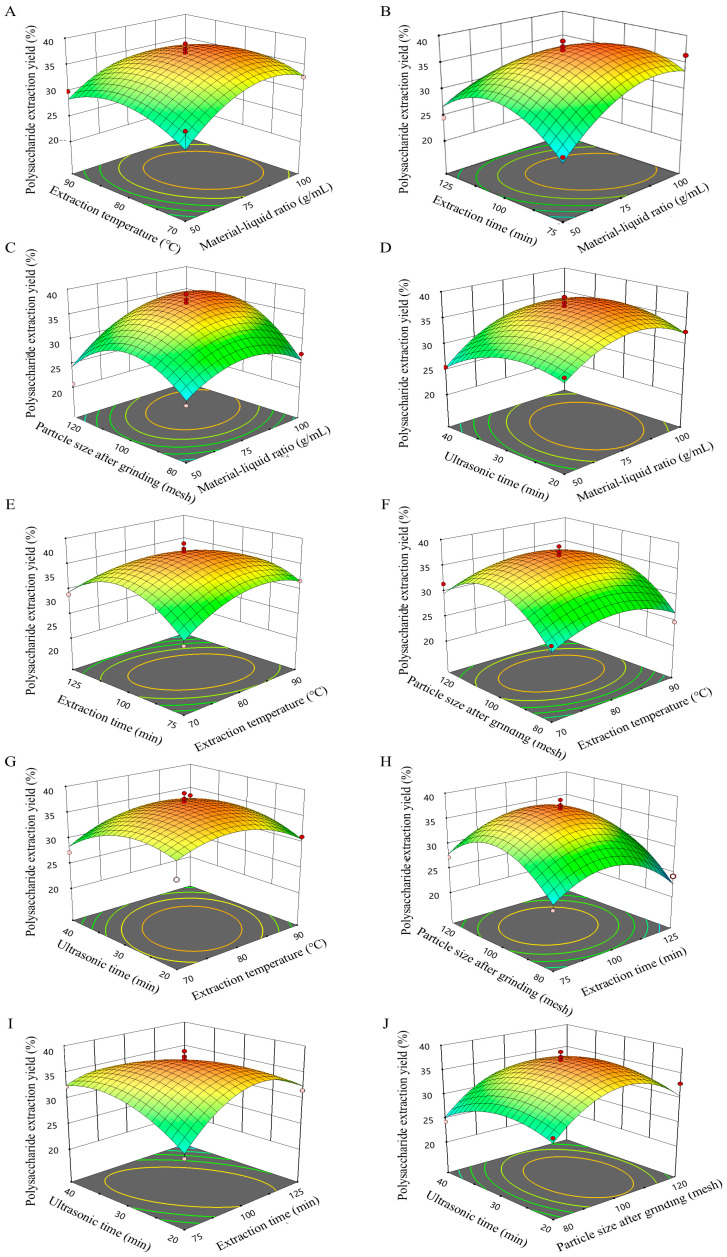
Response surface plots of the influence of any two factors on polysaccharide yield. Material–liquid and extraction temperature (**A**), material–liquid and extraction time (**B**), material–liquid and particle size after grinding (**C**), material–liquid and ultrasound time (**D**), extraction temperature and extraction time (**E**), extraction temperature and particle size after grinding (**F**), extraction temperature and ultrasound time (**G**), extraction time and particle size after grinding (**H**), extraction time and ultrasound time (**I**), particle size after grinding and ultrasound time (**J**) on the polysaccharide extraction rate, respectively.

**Figure 3 foods-13-03964-f003:**
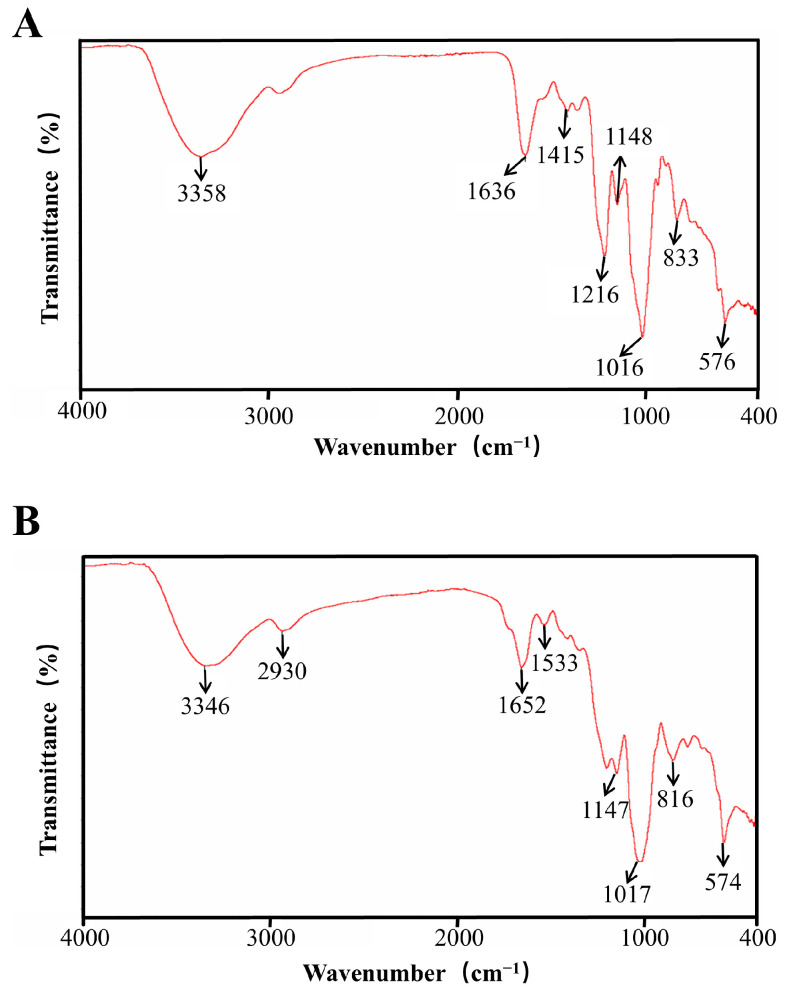
The FT-IR spectra of RIP-1 (**A**) and RIP-2 (**B**).

**Figure 4 foods-13-03964-f004:**
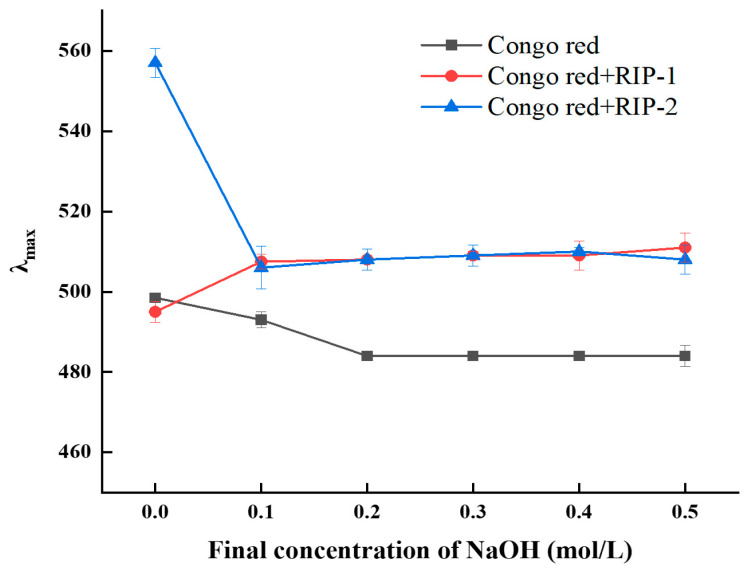
Analysis of spiral–curl transformation of RIP-1 and RIP-2 in different NaOH concentrations.

**Figure 5 foods-13-03964-f005:**
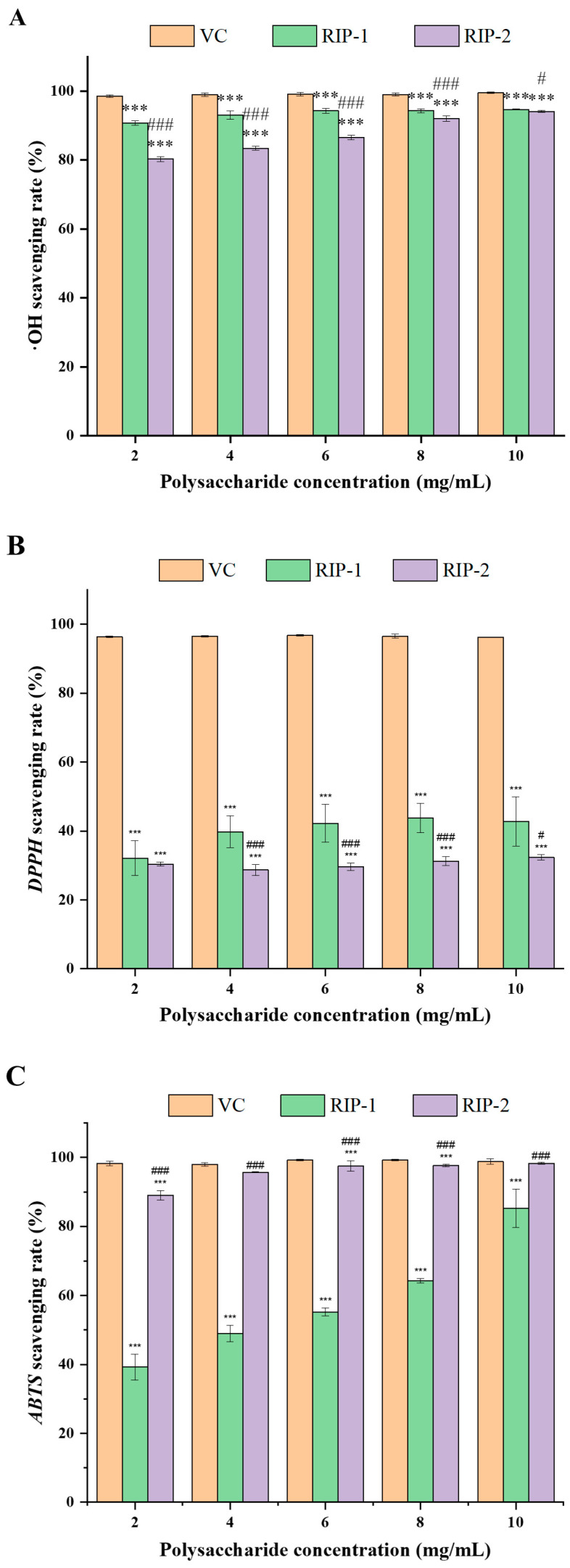
Bar chart of ·*OH* (**A**), *DPPH* (**B**), and *ABTS* scavenging effect (**C**) of polysaccharides from *Rhodymenia intricata* (mean ± SD, *n* = 3). (Compared with VC: *** *p* < 0.001; compared with RIP-1: # *p* < 0.05, ### *p* < 0.001).

**Table 1 foods-13-03964-t001:** The results of the steepest climbing test.

Serial Number	A (g/mL)	B (°C)	C (min)	D (mesh)	E (min)	Extraction Yield (%)
1	1:50	70	75	80	20	23.56
2	1:75	80	100	100	30	33.48
3	1:100	90	125	120	40	21.26

A, B, C, D, and E represent material–liquid ratio, temperature, time, particle size, and ultrasound time, respectively.

**Table 2 foods-13-03964-t002:** Box–Behnken design.

Factors	Unit	Symbols	Level of Factors		
			−1	0	1
Material-liquid ratio	g/mL	A	1:50	1:75	1:100
Extraction temperature	°C	B	70	80	90
Extraction time	min	C	75	100	125
Particle size after grinding	mesh	D	80	100	120
Ultrasonic time	min	E	20	30	40

**Table 3 foods-13-03964-t003:** The results of the Box–Behnken design.

Std. Order	A	B	C	D	E	Extraction Yield (%)
1	1:75	80	100	100	30	37.35
2	1:75	90	100	100	20	31.06
3	1:75	70	100	120	30	31.46
4	1:75	80	125	80	30	24.21
5	1:50	80	100	80	30	24.36
6	1:75	70	125	100	30	28.96
7	1:75	80	100	80	40	24.27
8	1:50	70	100	100	30	29.62
9	1:75	90	100	120	30	28.16
10	1:75	80	125	100	40	22.49
11	1:75	90	75	100	30	31.68
12	1:50	80	125	100	30	24.55
13	1:75	80	75	100	20	26.13
14	1:75	80	100	120	40	29.38
15	1:100	80	125	100	30	26.06
16	1:75	80	125	100	20	31.56
17	1:100	80	100	100	20	32.41
18	1:75	90	100	100	40	33.41
19	1:100	80	100	120	30	31.48
20	1:75	80	125	120	30	28.68
21	1:50	80	100	100	20	30.73
22	1:75	80	100	120	20	32.96
23	1:50	80	100	100	40	25.52
24	1:75	90	125	100	30	25.6
25	1:75	70	100	100	20	29.68
26	1:75	70	100	100	40	27.23
27	1:75	70	75	100	30	26.35
28	1:75	80	100	100	30	36.17
29	1:75	90	100	80	30	24.72
30	1:75	80	75	80	30	24.81
31	1:75	80	100	80	20	28.85
32	1:100	80	100	80	30	26.99
33	1:100	70	100	100	30	32.68
34	1:75	80	100	100	30	36.47
35	1:50	80	75	100	30	24.95
36	1:100	80	100	100	40	31.81
37	1:75	80	100	100	30	37.97
38	1:75	80	75	120	30	27.3
39	1:75	80	100	100	30	38.99
40	1:100	80	75	100	30	36.38
41	1:50	80	100	120	30	20.71
42	1:75	80	75	100	40	32.25
43	1:50	90	100	100	30	29.91
44	1:100	90	100	100	30	28.75
45	1:75	70	100	80	30	27.28
46	1:75	80	100	100	30	36.54

A, B, C, D, and E represent material–liquid ratio, temperature, time, particle size, and ultrasound time, respectively.

**Table 4 foods-13-03964-t004:** Results of the response surface polynomial models.

Variables	Sum of Squares	Degrees of Freedom	Mean Square	*F*-Value	*p*-Value	Significance
Model	776.88	20	38.84	9.39	<0.0001	**
A-A	81.95	1	81.95	19.81	0.0002	**
B-B	0.0001	1	0.0001	0	0.9971	
C-C	19.67	1	19.67	4.75	0.0389	*
D-D	37.95	1	37.95	9.17	0.0056	**
E-E	18.11	1	18.11	4.38	0.0468	*
AB	4.45	1	4.45	1.08	0.3095	
AC	24.6	1	24.6	5.95	0.0222	*
AD	16.56	1	16.56	4	0.0564	
AE	5.31	1	5.31	1.28	0.2679	
BC	18.88	1	18.88	4.56	0.0427	*
BD	0.1369	1	0.1369	0.0331	0.8571	
BE	5.76	1	5.76	1.39	0.2492	
CD	0.9801	1	0.9801	0.2369	0.6307	
CE	57.68	1	57.68	13.94	0.0010	**
DE	0.25	1	0.25	0.0604	0.8078	
A^2^	154.12	1	154.12	37.25	<0.0001	**
B^2^	100.83	1	100.83	24.37	<0.0001	**
C^2^	259.04	1	259.04	62.61	<0.0001	**
D^2^	311.72	1	311.72	75.34	<0.0001	**
E^2^	85.67	1	85.67	20.71	0.0001	**
Residual	103.44	25	4.14			
Lack of Fit	97.61	20	4.88	4.18	0.0596	
Pure Error	5.83	5	1.17			
Cor Total	880.32	45				
R^2^	0.8825					
Adjusted R^2^	0.7885					

A, B, C, D, and E represent material–liquid ratio, temperature, time, particle size, and ultrasound time, respectively; ** means *p* < 0.01, * means *p* < 0.05. The following annotations are all the same.

**Table 5 foods-13-03964-t005:** Results of deproteinization in relation to TCA concentration.

Percentage of TCA	Protein Removal Rate	Polysaccharide	Comprehensive Score
4%	38.95 ± 2.91	96.09 ± 1.18	67.52
6%	55.85 ± 0.73	92.94 ± 0.80	74.40
8%	62.61 ± 1.82	96.10 ± 1.60	79.36
10%	40.24 ± 7.97	93.77 ± 0.70	67.01
12%	44.74 ± 2.43	94.55 ± 1.07	69.30

**Table 6 foods-13-03964-t006:** Basic components of RIP-1 and RIP-2.

Sample	Total Sugars (%)	Proteins (%)	Sulfate Groups (%)	Polyphenols (%)
RIP-1	25.22 ± 0.28	23.29 ± 0.71	19.75 ± 1.17	0.29 ± 0.01
RIP-2	27.37 ± 0.54	10.01 ± 0.50	20.51 ± 3.85	0.29 ± 0.04

## Data Availability

The original contributions presented in the study are included in the article, further inquiries can be directed to the corresponding author.

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
