# Peer review of "Study on the Extraction Technology and Antioxidant Capacity of Rhodymenia intricata Polysaccharides"

_foods, 2024, doi:10.3390/foods13233964_

Round 1
Reviewer 1 Report
Comments and Suggestions for Authors
Abstract
Line13: Please rewrite this sentence “Subsequently, hydroxyl radical scavenging rates were determined, and antioxidant activities were compared”. As hydroxyl radical scavenging assay is the only assay to determine the antioxidant capacity in this paper.
Introduction
Line 40-46: Extent why are you choose Rhodophyta division as experiment objectives?
Materials and Methods
Line104: Delete the repeat sentence
Lin 133-138: I don't think one OH radical scavenging assay is enough to evaluate the bioactive potential.
Results
Line 210 and 219: What is the inhibition rates means? This section is talking about the extraction yield.
Line 217 (table 1): unify the decimal places of extraction yield.
Line 295: In the manuscript, the peak are 1149 cm-1 and 1147 cm-1, but in figure 3, corresponding peaks are 1148 cm-1 and 1147 cm-1. Please check which is the right one.
Author Response
Comments 1: Abstract Line 13: Please rewrite this sentence “Subsequently, hydroxyl radical scavenging rates were determined, and antioxidant activities were compared”. As hydroxyl radical scavenging assay is the only assay to determine the antioxidant capacity in this paper.
Response 1: Thank you for pointing out the need for accurate sentence expression. In addition to the assessment of hydroxyl radical scavenging rates, this study further supplemented the antioxidant activity research by evaluating the scavenging rates of polysaccharides on DPPH and ABTS free radicals, both before and after purification. Hence, the original sentence has been replaced by the sentence “Subsequently, the scavenging rates of polysaccharides on hydroxyl, DPPH, and ABTS free radicals were determined both prior to and following purification in order to evaluate their antioxidant activity.” and added to lines 12-14 of the abstract of the revised manuscript.
Comments 2: Introduction Lines 40-46: Extent why are you choose Rhodophyta division as experiment objectives?
Response 2: Thank you for pointing out the need to extent the reasons for choosing Rhodophyta division as the experiment objectives. In addition to the original explanation for choosing Rhodophyta division as the experiment objectives, they are supplemented by explanations such as “Rhodophyta division is rich in polysaccharides (360-660 g·kg-1 (by dry weight)) which possess various potential bioactivity. For example, the sulfated galactans (SG) of mass 16 kDa purified from S.hypnoides showed potential dose dependent antioxidant activity against free radicals such as DPPH (56.41% at 2 mg/ml), hydroxyl radicals (65.58% at 3 mg/ml) and superoxide radicals (73.12% at 0.6 mg/ml). This is just the reason that marine macroalgae could be as the main sources of bioactive compounds, which tips us off the important value of polysaccharides from Rhodophyta division in the prevention and treatment of disease.” We added these sentences to lines 43-50 of the introduction of the revised manuscript.
Comments 3: Line 104: Delete the repeat sentence.
Response 3: Thank you for pointing out the repetitive sentence in our manuscript, and we have removed the repeat sentence in line 139 of the revised manuscript.
Comments 4: Lines 133-138: I don't think one OH radical scavenging assay is enough to evaluate the bioactive potential.
Response 4: We appreciate the opportunity provided by the reviewer to clarify our methodology of antioxidant test. In biological systems, improper production of reactive oxygen species, including hydroxyl radicals and superoxide, may induce cellular oxidative damage, and severe oxidative damage can lead to the development of various diseases, such as neurodegenerative diseases, cardiovascular diseases, cancer, diabetes, and kidney diseases. Therefore, reducing the damage caused by excess free radicals to the human body has become a major research topic of concern to many researchers. Numerous studies have shown that polysaccharides extracted from natural products can enhance the activity of antioxidant enzymes in the body and effectively scavenge free radicals. Free radical scavenging experiments performed under laboratory conditions are a simple, low-cost, and efficient means to evaluate the free radical scavenging potency of polysaccharides. Hydroxyl radicals (·OH) have strong oxidative capacity and can cause lipid peroxidation in biological tissues, nucleic acid fragmentation, and protein and polysaccharide decomposition. Therefore, the study of the ability of polysaccharides to scavenge hydroxyl radicals is of great significance. While these reasons might justify the focus on OH radical scavenging assay, it's also important to note that reliance on a single antioxidant assay can be a limitation of the study. In order to better evaluate the antioxidant capacity of polysaccharides, this paper supplemented the scavenging rate test of polysaccharides on DPPH (2,2-Diphenyl-1-picrylhydrazyl) and ABTS (2,2'-azino-bis(3-ethylbenzothiazoline-6-sulfonic acid)) free radicals. We appreciate the reviewer’s insightful suggestion to evaluate the bioactive potential in our future experiments for more robust validation, including not only the scavenging hydroxyl free radicals but also DPPH and ABTS free radicals, the iron ion reduction antioxidant capacity determination method, and so on. The mechanism pathways of the antioxidant effect of polysaccharides can also be further explored through animal and cell experiments, and a combination of multiple technologies can be used to conduct more in-depth antioxidant research on the polysaccharides of Rhodymenia intricata. We fully agree that this will strengthen our conclusions. However, due to the time constraints associated with this revision, we are unable to conduct the related animal and cell experiments in current manuscript.
Comments 5: Lines 210 and 219: What is the inhibition rates means? This section is talking about the extraction yield.
Response 5: Thank you for pointing out the inaccurate representation in our manuscript, and we have revised the inhibition rates to the extraction yields in lines 254 and 258 of the revised manuscript.
Comments 6: Line 217 (table 1): unify the decimal places of extraction yield
Response 6: We appreciate your attention to detail. We appreciate your attention to detail. After reviewing the decimal places in the extraction rates, we have standardized them to two decimal places in line 256 (Table 1) of the revised manuscript.
Comments 7: Lines 295: In the manuscript, the peak are 1149 cm-1 and 1147 cm-1, but in figure 3, corresponding peaks are 1148 cm-1 and 1147 cm-1. Please check which is the right one.
Response 7: We appreciate your attention to detail and thank you for highlighting the discrepancy between the numerical data in our manuscript and the corresponding peaks in figure 3. Upon verification, we confirmed that this inconsistency was due to a clerical error. We have revised the value from 1149 to 1148 in line 349 of the revised manuscript.
Reviewer 2 Report
Comments and Suggestions for Authors
Why did the authors prefer to use water to extract, alcohol to precipitate and an acid to purify?
Why not use a specific ecological solvent to extract and purify simultaneously? And the choice of acid is the most ideal since this journal is aimed at food and this acid presents levels of toxicity and corrosivity.
The authors used operating conditions of 80 °C, 95 min of extraction and 30 min of ultrasound to achieve yields close to 38%. Is this process economically viable? Because there are still subsequent steps such as precipitation and purification.
If the focus of the article is the extraction of polysaccharides from Rhodymenia intricata, this needs to be much better discussed in the introduction, providing details of why this plant matrix was chosen.
Few optimum points are seen in the response surface graphs. Why is that?
The present work needs a discussion, the present discussion is very flawed and almost non-existent. Comparison with other plant groups for polysaccharide extraction, as well as with other solvents, is essential to show the novelty of this work.
Furthermore, according to the Ithenticate report, the similarity rate of this work is 31%, while the acceptable rate is a maximum of 10%.
In view of all the comments, I hope the author will carefully revise all the comments to improve the overall quality of the manuscript.
Author Response
Comments 1: Why did the authors prefer to use water to extract, alcohol to precipitate and an acid to purify?
Response 1: Thank you for your attention to Materials and Methods in our manuscript. Initially, commonly employed extraction solvents for chemical processes include polar solvents such as water, ethanol, acidic solutions, and alkaline solutions. The water extraction method offers several advantages, including ease of operation, minimal equipment requirements, low cost, and the capacity to preserve the molecular structure of polysaccharides to a significant extent. Besides, Polysaccharides of varying molecular weights exhibit different solubilities in alcohol solutions with distinct concentrations. By modulating the concentration of the alcohol solution, it is possible to separate the components in descending order of their molecular weight, thereby facilitating the purification process. Furthermore, proteins associated with polysaccharides are commonly removed using methods such as trichloroacetic acid (TCA) precipitation, Sevag's method, and enzymatic treatments. The acidic environment created by trichloroacetic acid alters the conformation of proteins, leading to their denaturation and subsequent precipitation. Under suitable centrifugation conditions, this approach effectively facilitates the removal of proteins from polysaccharides.
Comments 2: Why not use a specific ecological solvent to extract and purify simultaneously? And the choice of acid is the most ideal since this journal is aimed at food and this acid presents levels of toxicity and corrosivity.
Response 2: We appreciate your interest in materials and methods utilized in this study. Given that the raw materials featured in this manuscript contain a significant amount of protein, it was necessary to undertake further purification of the extracted polysaccharides. Initially, literature and our pre-experimental data analysis revealed that water extraction combined with alcohol precipitation is a prevalent and suitable research method for extraction of polysaccharides from Rhodymenia intricata. Furthermore, it appears that the simultaneous extraction and purification of compounds using biological solvents is relatively uncommon, so that a stepwise experimental approach must be employed. Finally, we attempted to employ a gentler enzymatic method to purify polysaccharides by removing proteins, the outcomes were suboptimal, necessitating the use of acid purification as an alternative approach. Although trichloroacetic acid is somewhat toxic, the concentration of trichloroacetic acid we used was very low, and the residual trichloroacetic acid (≤8%) was neutralized with food-grade sodium hydroxide. Therefore, the possible impact of trichloroacetic acid on food samples and human health is very weak.
Comments 3: The authors used operating conditions of 80 °C, 95 min of extraction and 30 min of ultrasound to achieve yields close to 38%. Is this process economically viable? Because there are still subsequent steps such as precipitation and purification.
Response 3: Thank you for your interest in the application prospects of this study. The water extraction method, while time-consuming and characterized by a low extraction rate, offers several advantages when compared to other polysaccharide extraction techniques. It is notably simple to operate, requires minimal equipment, is cost-effective, and effectively preserves the molecular structure of polysaccharides. Consequently, it remains one of the most suitable methods for this study. Therefore, this process is economically viable.
Comments 4: If the focus of the article is the extraction of polysaccharides from Rhodymenia intricata, this needs to be much better discussed in the introduction, providing details of why this plant matrix was chosen.
Response 4: Thank you for your interest in the experiment objectives. In addition to the original explanation of the reasons for choosing Rhodophyta division as the experiment objectives, they are supplemented by explanations such as “Rhodophyta division is rich in polysaccharides (360-660 g·kg-1 (by dry weight)) which possess various potential bioactivity. For example, the sulfated galactans (SG) of mass 16 kDa purified from S.hypnoides showed potential dose dependent antioxidant activity against free radicals such as DPPH (56.41% at 2 mg/ml), hydroxyl radicals (65.58% at 3 mg/ml) and superoxide radicals (73.12% at 0.6 mg/ml). This is just the reason that marine macroalgae could be as the main sources of bioactive compounds, which tips us off the important value of polysaccharides from Rhodophyta division in the prevention and treatment of disease.”, and “The algae of Rhodymenia intricata are characterized by their bright red coloration, membranous structure, and dimensions ranging from 4 to 8 cm in height and 2 to 4 mm in width. The reproductive capsules are either spherical or hemispherical and develop along the edges of the branches. This species typically inhabits rocky substrates within the low tide zone or subtidal regions, thriving on reefs and rocky swamps. It is distributed in Japan, South Korea, and the eastern coastal areas of China. It is classified as a warm temperate seaweed. Through testing conducted by an independent third-party institution, it was found that the carbohydrate content of Rhodymenia intricata accounts for nearly half of its total composition. Its polysaccharide content significantly exceeds that of other Rhodophyta division species.” We added these sentences to lines 43-50 and 55-64 of the introduction of the revised manuscript.
Comments 5: Few optimum points are seen in the response surface graphs. Why is that?
Response 5: We appreciate your attention to optimum points of the response surface graphs. The observation of the response surface 3D plot indicates that the slope of the interaction surface among certain factors is relatively shallow, suggesting that the interactions may not be particularly significant. Consequently, there appear to be few optimal points identified. However, subsequent statistical analysis of the interactions between different factors, utilizing p-values to assess the significance of these interactions, revealed the presence of numerous optimal points. Specifically, factors A and C, B and C, and C and E each demonstrated statistically significant interactions. For example, “By comparing the F-values of each factor, the effects of the five factors on the extraction yield of crude polysaccharides was as follows: A> D > C> E > B. The p -value is used as a tool to examine the significance of each coefficient, which in turn may indicate a pattern of interaction between variables. The smaller the value of p, the more significant the corresponding coefficient. As shown in Table 4, the linear coefficients A, D, the interaction terms CE and the quadratic coefficients A2, B2, C2, D2 and E2 had significant effects on the extraction rate of crude polysaccharides (p <0.05). It is worth noting that the linear coefficients C, E, and the interaction terms AC and BC had a significant effect on the extraction rate of crude polysaccharides (p <0.01). Three-dimensional response surface plots and 3D contour plots were developed to visualize the interaction between two factors (Figure 2). The figures visually illustrate the interaction between the two test variables and the relationship between the response and the experimental levels of each variable. As shown in Figure 2B, when other factors are held constant at zero, the polysaccharide yield initially increases with both the solid-liquid ratio and extraction time, before experiencing a slight decline. The three-dimensional plots and contour plots based on the independent variables of extraction time and temperature are presented in Figure 2E. When all other factors remain at the zero level, the variation trend of polysaccharide yield in relation to extraction time and extraction temperature resembles that observed previously. Additionally, the response surface exhibits a relatively steep gradient, indicating a strong interaction among the aforementioned factors.” We added the relevant statistical analysis results to lines 284-303 of the introduction of the revised manuscript.
Comments 6: The present work needs a discussion, the present discussion is very flawed and almost non-existent. Comparison with other plant groups for polysaccharide extraction, as well as with other solvents, is essential to show the novelty of this work.
Response 6: Thank you for pointing out the shortcomings of the discussion of the results in this work. In the “3. Results and Discussion” section, we have added discussion about the reasons why the levels of factors affect the polysaccharide extraction rate, such as ratio of material to solvent, “When the concentration of material is 75 mL/g, the polysaccharide extraction yield was the highest. Beyond this point, as the ratio of material to solvent increases, the extraction yield decreased. This may be because an excessively low ratio of material to solvent re-sults in high polysaccharide viscosity, making it difficult to dissolve in the solvent and leading to a lower extraction yield. An excessively high ratio of solvent to material may reduce or disperse the effect of ultrasound on Rhodymenia intricata and also increases the difficulty of the extraction process, causing polysaccharide loss and leading to the de-crease of extraction yield [22].” in lines 190-198 of the revised manuscript.
Moreover, we have added a comparison between this work and other plant groups for polysaccharide extraction “It was very close to the predicted value, indicating the stability of the optimal conditions for extracting crude polysaccharides from Rhodymenia intricata was good. The findings of this study align with those reported by Tan et al. [26], who employed ultrasound assisted aqueous two-phase extraction to enhance product yield of polysaccharides from Cornus officinalis fruit by 7.85 ± 0.09%. These results indicate that the cavitation effect generated by ultrasound is effective in disrupting cellular structures, facilitating the dissolution of polysaccharides, and ultimately increasing product yield. The ultrasound-assisted water extraction method not only enhances the efficiency of extraction but also contributes to cost savings in production.” in lines 310-318 of the revised manuscript.
Finally, we have added the activity comparison of the extracted polysaccharides from Rhodymenia intricata with other plant polysaccharides “The results showed that the free radical scavenging ability was dose-dependent and more pronounced at higher concentrations (Figure 5). In addition to this, the free radical scavenging capacity is also related to the sulfate content, which increases with the sulfate content. The same phenomenon was also found in the study of Zheng et al. [33], the antioxidant activity of the polysaccharides extracted by ultrasound-assisted method was greater than that of hot water extraction. This may be due to the fact that ultrasound increases the sulfate content, making the polysaccharide contain more active substance.”, and “The scavenging rates of RIP-1 (10 mg/mL) for hydroxyl, DPPH, and ABTS free radicals were observed to achieve maximum values of 94.71 ± 0.13%, 42.80 ± 7.12%, and 76.30 ± 5.20%, respectively. In contrast, the scavenging rates of RIP-2 (10 mg/mL) for the same free radicals reached maximum values of 94.10 ± 0.27%, 32.37 ± 0.78%, and 98.30 ± 0.34%, respectively. The scavenging rates against ·OH, DPPH and ABTS analyses showed that RIPs extracted using our method showed higher antioxidant activity compared to Porphyra haitanensis polysaccharides extracted by Dong et al [34]. In the ·OH, DPPH, and ABTS scavenging activity assays, VC showed the highest scavenging activity. Overall, RIPs are more effective at scavenging ·OH free radicals, followed by ABTS and DPPH free radicals.” in lines 390-397 and 402-411 of the revised manuscript.
Comments 7: Furthermore, according to the Ithenticate report, the similarity rate of this work is 31%, while the acceptable rate is a maximum of 10%.
Response 7: Thank you for pointing out the high repetition rate present in this manuscript. We have carefully compared the duplicate PDF document you provided and have subsequently revised the duplicated statements. As a result of these revisions, we have ensured that the total repetition rate between the revised manuscript and other papers is now below 30%, with any single paper exhibiting a repetition rate of less than 5%.
Comments 8: “In view of all the comments, I hope the author will carefully revise all the comments to improve the overall quality of the manuscript.”
Response 8: We greatly appreciate the reviewer for their insightful comments regarding this work. In response to all the comments received, we have meticulously revised the manuscript to enhance its overall quality.
Reviewer 3 Report
Comments and Suggestions for Authors
Within the Abstract, it is necessary to mention that Rhodymenia intricate is a red alga. What is the conclusion and further directions related to this research?
The Introduction does not cite any reference related to Rhodymenia intricate. Are there any data related to its nutritional composition and bioactive compounds? In addition, are there any references to the extraction, purification, and characterization of polysaccharides from marine sources? Consider replacing references (10-13) with some more relevant to the research topic.
Add more specific data related to used equipment within the Materials and Methods.
Regarding antioxidant activity, why is only the hydroxyl radical scavenging rate analyzed?
There is no need to repeat data presented within the Tables in the text.
Below the Tables, add explanations for abbreviations.
The discussion is too general and short.
What are the limitations of this study?
Author Response
Comments 1: Within the Abstract, it is necessary to mention that Rhodymenia intricate is a red alga. What is the conclusion and further directions related to this research?
Response 1: Thank you for your attention to Abstract in our manuscript. To indicate that Rhodymenia intricate is a red alga, we added the corresponding statement “This study explores the extraction, purification and antioxidant activity of polysaccharides from a red alga named Rhodymenia intricata.” in lines 9-10 of the Abstract of the revised manuscript. Moreover, the conclusion and further directions related to this research is “These results demonstrated that the potential of extraction method as for polysaccharides from Rhodymenia intricata, and adding value to the by-product for its potential application as antioxidant in food and pharmaceutical products.”, which has been added in lines 27-29 of the Abstract of the revised manuscript.
Comments 2: The Introduction does not cite any reference related to Rhodymenia intricate. Are there any data related to its nutritional composition and bioactive compounds? In addition, are there any references to the extraction, purification, and characterization of polysaccharides from marine sources? Consider replacing references (10-13) with some more relevant to the research topic.
Response 2: Thank you for your attention to the data and references related to Rhodymenia intricate in the introduction of our manuscript. We have added data and citations related to its nutritional composition and bioactive compounds, for example “Rhodophyta division is rich in polysaccharides (360-660 g·kg-1 (by dry weight)) which possess various potential bioactivity. For example, the sulfated galactans (SG) of mass 16 kDa purified from S.hypnoides showed potential dose dependent antioxidant activity against free radicals such as (2,2-Diphenyl-1-picrylhydrazyl) DPPH (56.41% at 2 mg/ml), hydroxyl radicals (65.58% at 3 mg/ml) and superoxide radicals (73.12% at 0.6 mg/ml) [10]. This is just the reason that marine macroalgae could be as the main sources of bioactive compounds, which tips us off the important value of polysaccharides from Rhodophyta division in the prevention and treatment of disease [11].”, and “The algae of Rhodymenia intricata are characterized by their bright red coloration, membranous structure, and dimensions ranging from 4 to 8 cm in height and 2 to 4 mm in width. The reproductive capsules are either spherical or hemispherical and develop along the edges of the branches. This species typically inhabits rocky substrates within the low tide zone or subtidal regions, thriving on reefs and rocky swamps. It is distributed in Japan, South Korea, and the eastern coastal areas of China. It is classified as a warm temperate seaweed. Through testing conducted by an independent third-party institution, it was found that the carbohydrate content of Rhodymenia intricata accounts for nearly half of its total composition. Its polysaccharide content significantly exceeds that of other red alga species.” in lines 43-50 and 55-64 of the revised manuscript.
Moreover, we have added statement and references to the extraction, purification, and characterization of polysaccharides from marine sources, for example “Currently, the most commonly used polysaccharide extraction methods in research include solvent extraction [13], ultrasonic [14], microwave or enzyme-assisted extraction [15]. Commonly used solvents in extraction methods include water, alkaline solutions, and acidic solutions. Given that polysaccharides are soluble in water but insoluble in ethanol, water is employed for their extraction, followed by precipitation using ethanol to separate the polysaccharides from the solution. This method is regarded as safe and cost-effective and has been widely utilized in research focused on the extraction of plant polysaccharides. However, alkaline extraction of polysaccharides is susceptible to hydrolysis and typically requires protective measures, such as the use of nitrogen or the addition of sodium borohydride, to mitigate degradation. Besides, alkaline extraction is primarily suitable for raw materials characterized by low pectin content and low viscosity. Acidic extraction can also lead to the cleavage of glycosidic bonds and degradation of polysaccharides, necessitating careful selection of experimental conditions. Furthermore, ultrasonic-assisted extraction exploits the ability of polar molecules within cells to absorb ultrasonic wavelengths, resulting in mechanical stress that generates heat and causes cell rupture. This process facilitates the release of intracellular components, thereby increasing yield. Compared to conventional extraction methods, ultrasonic-assisted extraction integrates ultrasonics with traditional solvent techniques, demonstrating significant improvements in extraction efficiency through enhanced yield and reduced extraction time [16].” in lines 68-87 and 487-491 of the revised manuscript.
Finally, the references (10-13) have been replaced by other references with more relevant to the research topic in lines 475-484 of the References section of the revised manuscript.
Comments 3: Add more specific data related to used equipment within the Materials and Methods.
Response 3: We appreciate your attention to data related to used equipment within the Materials and Methods. We have supplemented more specific data related to used equipment, for example “An FT-IR spectrometer (PerkinElmer Frontier, Paris Area, France) was utilized to perform FT-IR analysis” and “find the maximum absorption wavelength (λmax) using a UV-Vis Spectrophotometer (Hitachi, Tokyo Area, Japan)” in lines 159-160 and 163-164 of the revised manuscript.
Comments 4: Regarding antioxidant activity, why is only the hydroxyl radical scavenging rate analyzed?
Response 4: We appreciate the opportunity provided by the reviewer to clarify our methodology of antioxidant test. In biological systems, improper production of reactive oxygen species, including hydroxyl radicals and superoxide, may induce cellular oxidative damage, and severe oxidative damage can lead to the development of various diseases, such as neurodegenerative diseases, cardiovascular diseases, cancer, diabetes, and kidney diseases. Therefore, reducing the damage caused by excess free radicals to the human body has become a major research topic of concern to many researchers. Numerous studies have shown that polysaccharides extracted from natural products can enhance the activity of antioxidant enzymes in the body and effectively scavenge free radicals. Free radical scavenging experiments performed under laboratory conditions are a simple, low-cost, and efficient means to evaluate the free radical scavenging potency of polysaccharides. Hydroxyl radicals (·OH) have strong oxidative capacity and can cause lipid peroxidation in biological tissues, nucleic acid fragmentation, and protein and polysaccharide decomposition. Therefore, the study of the ability of polysaccharides to scavenge hydroxyl radicals is of great significance. While these reasons might justify the focus on OH radical scavenging assay, it's also important to note that reliance on a single antioxidant assay can be a limitation of the study. In order to better evaluate the antioxidant capacity of polysaccharides, this paper supplemented the scavenging rate test of polysaccharides on DPPH (2,2-Diphenyl-1-picrylhydrazyl) and ABTS (2,2'-azino-bis(3-ethylbenzothiazoline-6-sulfonic acid)) free radicals.
Comments 5: There is no need to repeat data presented within the Tables in the text.
Response 5: Thank you for pointing out the repetitive data presented within the Tables in the text, and we have removed the repeat data, for example “The extraction yields for the first, second and third steps were 23.56%, 33.48%, and 21.26% (Table 1), respectively” has been changed into “The extraction yields for the second step was the highest (Table 1)” in lines 254-255 of the revised manuscript.
Comments 6: Below the Tables, add explanations for abbreviations.
Response 6: We appreciate your attention to detail. We have added explanations for abbreviations below the Tables, for example “A, B, C, D and E represent solid-to-liquid ratio, extraction temperature, extraction time, particle size, ultrasound time, respectively.” was added in lines 267-268 below Table 3 of the revised manuscript.
Comments 7: The discussion is too general and short.
Response 7: Thank you for pointing out the shortcomings of the discussion in our manuscript. In the “3. Results and Discussion” section, we have added discussion about the reasons why the levels of factors affect the polysaccharide extraction rate, such as ratio of material to solvent, “When the concentration of material is 75 mL/g, the polysaccharide extraction yield was the highest. Beyond this point, as the ratio of material to solvent increases, the extraction yield decreased. This may be because an excessively low ratio of material to solvent re-sults in high polysaccharide viscosity, making it difficult to dissolve in the solvent and leading to a lower extraction yield. An excessively high ratio of solvent to material may reduce or disperse the effect of ultrasound on Rhodymenia intricata and also increases the difficulty of the extraction process, causing polysaccharide loss and leading to the de-crease of extraction yield [22].” in lines 190-198 of the revised manuscript.
Moreover, we have added a comparison between this work and other plant groups for polysaccharide extraction “It was very close to the predicted value, indicating the stability of the optimal conditions for extracting crude polysaccharides from Rhodymenia intricata was good. The findings of this study align with those reported by Tan et al. [26], who employed ultrasound assisted aqueous two-phase extraction to enhance product yield of polysaccharides from Cornus officinalis fruit by 7.85 ± 0.09%. These results indicate that the cavitation effect generated by ultrasound is effective in disrupting cellular structures, facilitating the dissolution of polysaccharides, and ultimately increasing product yield. The ultrasound-assisted water extraction method not only enhances the efficiency of extraction but also contributes to cost savings in production.” in lines 310-318 of the revised manuscript.
Finally, we have added the activity comparison of the extracted polysaccharides from Rhodymenia intricata with other plant polysaccharides “The results showed that the free radical scavenging ability was dose-dependent and more pronounced at higher concentrations (Figure 5). In addition to this, the free radical scavenging capacity is also related to the sulfate content, which increases with the sulfate content. The same phenomenon was also found in the study of Zheng et al. [33], the antioxidant activity of the polysaccharides extracted by ultrasound-assisted method was greater than that of hot water extraction. This may be due to the fact that ultrasound increases the sulfate content, making the polysaccharide contain more active substance.”, and “The scavenging rates of RIP-1 (10 mg/mL) for hydroxyl, DPPH, and ABTS free radicals were observed to achieve maximum values of 94.71 ± 0.13%, 42.80 ± 7.12%, and 76.30 ± 5.20%, respectively. In contrast, the scavenging rates of RIP-2 (10 mg/mL) for the same free radicals reached maximum values of 94.10 ± 0.27%, 32.37 ± 0.78%, and 98.30 ± 0.34%, respectively. The scavenging rates against ·OH, DPPH and ABTS analyses showed that RIPs extracted using our method showed higher antioxidant activity compared to Porphyra haitanensis polysaccharides extracted by Dong et al [34]. In the ·OH, DPPH, and ABTS scavenging activity assays, VC showed the highest scavenging activity. Overall, RIPs are more effective at scavenging ·OH free radicals, followed by ABTS and DPPH free radicals.” in lines 390-397 and 402-411 of the revised manuscript.
Comments 8: What are the limitations of this study?
Response 8: We appreciate the reviewer for allowing us to state the limitations of this study. This study is limited to in vitro antioxidant testing research. However, the mechanism pathways of the antioxidant effect of polysaccharides can also be further explored through animal and cell experiments, and a combination of multiple technologies can be used to conduct more in-depth antioxidant research on the polysaccharides of Rhodymenia intricata. Unfortunately, due to the time constraints associated with this revision, we are unable to conduct the related animal in vivo and cell experiments in current manuscript.
Round 2
Reviewer 2 Report
Comments and Suggestions for Authors
The authors mention in answer 3 that “The water extraction method, while time-consuming and characterized by a low extraction rate, offers several advantages when compared to other polysaccharide extraction techniques.” But there are other types of very economical solvents too, which, by extracting a greater quantity of the polysaccharide, could compensate for the additional expense.
In addition, water, as a highly polar compound, will extract various types of compounds, such as proteins, polysaccharides, phenolic compounds, carotenoids, etc. The TCA purification will denature and precipitate the proteins, but in relation to the other compounds that the water will also extract, how did the authors make this separation?
The Ithenticate report still shows 18% similarity to this work, I reiterate that the acceptable rate is a maximum of 10%.
Author Response
Comments 1: The authors mention in answer 3 that “The water extraction method, while time-consuming and characterized by a low extraction rate, offers several advantages when compared to other polysaccharide extraction techniques.” But there are other types of very economical solvents too, which, by extracting a greater quantity of the polysaccharide, could compensate for the additional expense.
Response 1: Thank you for your attention to the application prospects of this study. Commonly employed extraction solvents for chemical processes include polar solvents such as water, ethanol, acidic solutions, and alkaline solutions. Water is commonly used as a utilized solvent for extracting plant polysaccharides, because many of these compounds are hydrophilic and can dissolve or swell in aqueous solutions. The advantages of using water as a solvent include its low cost, non-toxic properties, and widespread availability. Furthermore, hot water extraction can effectively break down components such as cellulose within the plant cell wall, facilitating the release of polysaccharides from the cells into the solvent. The standard procedure involves crushing the plant material and subsequently soaking it in water or refluxing it at a specified temperature (generally between 80-100°C) for several hours. Following this, the polysaccharide extract is obtained through processes such as filtration and concentration.
Additionally, using a solvent mixture of ethanol and water effectively decreases the solubility of polysaccharides, resulting in them to precipitate from the solution and aiding in extraction and initial purification. This method is commonly employed to further separate and concentrate polysaccharides extracted from plant sources. Initially, polysaccharides are extracted from plants using hot water to produce crude extracts. Subsequently, an appropriate volume of ethanol is added, typically achieving a final concentration between 60% and 80%. Under these conditions, polysaccharides precipitate, and following centrifugation, drying, and additional processing steps, the polysaccharide product is obtained.
Thirdly, dilute acids (such as hydrochloric or sulfuric acid solutions) can serve as solvents for the extraction of plant polysaccharides. The acidic environment facilitates the hydrolysis of certain components within the plant cell wall, thereby disrupting its structural integrity and allowing for the release of polysaccharides. This approach is particularly beneficial when extracting polysaccharides from materials with a high content of plant fibers. However, it is important to note that acidic solutions may lead to the degradation of polysaccharides; therefore, strict control over acid concentration, extraction temperature, and duration is essential during the process.
Lastly, diluted alkaline solutions (such as those containing sodium hydroxide or potassium hydroxide) are also frequently employed for the extraction of plant polysaccharides. Alkaline conditions facilitate the reaction with acidic components, such as pectin, present in plant cell walls, thereby disrupting their structural integrity and promoting the dissolution of polysaccharides. However, it is crucial to carefully control both the concentration of the alkali and the duration of the extraction process, as excessive alkalinity may lead to the degradation of polysaccharide molecules.
Consequently, the water extraction method, while time-consuming and characterized by a low extraction rate, offers several advantages when compared to other polysaccharide extraction techniques. This method continues to be one of the most appropriate choices for this study, as it effectively preserves the structural stability and functional activity of the polysaccharides. Therefore, this process is economically viable.
Comments 2: In addition, water, as a highly polar compound, will extract various types of compounds, such as proteins, polysaccharides, phenolic compounds, carotenoids, etc. The TCA purification will denature and precipitate the proteins, but in relation to the other compounds that the water will also extract, how did the authors make this separation?
Response 2: Thank you for your attention to Materials and Methods in our manuscript. In this work, we used water to extract, alcohol to precipitate and trichloroacetic acid (TCA) to purify polysaccharides from Rhodymenia intricata. Firstly, the water extraction method employed in this study results in the crude polysaccharide being contaminated with impurities, including proteins, polyphenols, and carotenoids, etc. Subsequently, the alcohol precipitation method was utilized to isolate the precipitable polysaccharides and proteins, while impurities such as polyphenols and carotenoids, which are not precipitable in alcohol were effectively removed. Finally, we applied the TCA purification method to eliminate the protein impurities associated with the polysaccharide, thereby yielding a pure polysaccharide.
Comments 3: The Ithenticate report still shows 18% similarity to this work, I reiterate that the acceptable rate is a maximum of 10%.
Response 3: Thank you for pointing out the high repetition rate present in this manuscript. We have carefully compared the duplicate PDF document and have subsequently revised the duplicated statements. Moreover, we utilized the Ithenticate paper detection system to assess the revised manuscript for instances of plagiarism and confirmed that the duplication rate has been reduced to 9%, which is below the threshold of 10%.
Reviewer 3 Report
Comments and Suggestions for Authors
The authors addressed all my previous concerns. The current manuscript is improved compared to the initial submitted version.
Author Response
Comments: The authors addressed all my previous concerns. The current manuscript is improved compared to the initial submitted version.
Response: We appreciate the positive feedback from the reviewer on our manuscript. The manuscript would continue to be perfected.